# The Role of Obesity-Induced Perivascular Adipose Tissue (PVAT) Dysfunction in Vascular Homeostasis

**DOI:** 10.3390/nu13113843

**Published:** 2021-10-28

**Authors:** Agata Stanek, Klaudia Brożyna-Tkaczyk, Wojciech Myśliński

**Affiliations:** 1Department and Clinic of Internal Medicine, Angiology and Physical Medicine, Faculty of Medical Sciences in Zabrze, Medical University of Silesia, Batorego 15 Street, 41-902 Bytom, Poland; 2Chair and Department of Internal Medicine, Medical University of Lublin, Staszica 16 Street, 20-081 Lublin, Poland; klaudiabrozyna19@gmail.com (K.B.-T.); wojciech.myslinski@umlub.pl (W.M.)

**Keywords:** obesity, perivascular adipose tissue, exercise, endothelial dysfunction

## Abstract

Perivascular adipose tissue (PVAT) is an additional special type of adipose tissue surrounding blood vessels. Under physiological conditions, PVAT plays a significant role in regulation of vascular tone, intravascular thermoregulation, and vascular smooth muscle cell (VSMC) proliferation. PVAT is responsible for releasing adipocytes-derived relaxing factors (ADRF) and perivascular-derived relaxing factors (PDRF), which have anticontractile properties. Obesity induces increased oxidative stress, an inflammatory state, and hypoxia, which contribute to PVAT dysfunction. The exact mechanism of vascular dysfunction in obesity is still not well clarified; however, there are some pathways such as renin–angiotensin–aldosterone system (RAAS) disorders and PVAT-derived factor dysregulation, which are involved in hypertension and endothelial dysfunction development. Physical activity has a beneficial effect on PVAT function among obese patients by reducing the oxidative stress and inflammatory state. Diet, which is the second most beneficial non-invasive strategy in obesity treatment, may have a positive impact on PVAT-derived factors and may restore the balance in their concentration.

## 1. Introduction

Today, an increasing prevalence of obesity is observed in many countries, since a third of the worldwide population is described as obese or overweight [1]. National survey data from 2000 to 2018 in the USA reported that obesity prevalence increased to over 42% among adults, and the prevalence of severe obesity (BMI ≥ 40 kg/m^2^) doubled to 9.2% over the study period [2]. Weight problems and obesity are increasing at a rapid rate in most of the EU Member States, with estimates of 52.7% of the EU’s population being overweight in 2019 [3]. According to the Global Burden of Disease study, 4.7 million people died prematurely in 2017 as a result of obesity [4]. Obesity is a risk factor for developing many disorders such as diabetes mellitus, hypertension, cardiovascular events, obstructive sleep apnea syndrome, certain cancers, and musculoskeletal diseases [5]. Obesity also has a negative impact on quality of life and increases the costs of healthcare [6,7].

Adipose tissue is known as an endocrine organ. By producing adipokines, it regulates various metabolism pathways and processes such as insulin sensitivity, energy metabolism, blood flow, and even inflammatory stage [8,9]. Adipose tissue is divided into two main subtypes: white (WAT) and brown (BAT), according to their characteristic and different properties. WAT is responsible for storage of the excess of energy as fatty acids, while BAT mostly specializes in thermogenesis [10]. There is also a third type of adipocyte, termed the “beige” adipocyte. It is a brown adipocyte that arises within white adipose depots and also has thermogenic capacity [11].

Perivascular adipose tissue (PVAT) is an additional special type of adipose tissue surrounding blood vessels. PVAT is located around the aorta, coronary arteries, small and resistance vessels, and vasculature of the musculoskeletal system [12,13,14] On the contrary, PVAT is absent among cerebral vessels [15]. It consists of stem cells, adipocytes, mast cells, and nerves [16,17]. PVAT lies outside the adventitia, with no laminar structures or any organized barrier separating them from each other. Although PVAT characteristics resemble both brown and white adipose tissues, recent evidence suggests that PVAT develops from its own distinct precursors, implying a closer link between PVAT and the vascular system [18]. PVAT at different anatomical locations presents different phenotypes. PVAT demonstrates WAT, BAT, and mixed phenotypes, depending on their anatomical placement [19]. In the abdominal PVAT, white adipocytes are more abundant, whereas thoracic PVAT contains more brown adipocytes. These regional differences in PVAT could explain the higher susceptibility of the abdominal aorta to atherosclerosis compared to the thoracic aorta [20,21,22]. Moreover, gender influences differences in PVAT. After menopause in women, there is an increase in perivascular and pericardial adipose tissue, and additionally, the volume of aortic PVAT positively correlates with the reduction in estradiol [23,24]. Additionally, in obesity experimental models, the PVAT mass and adipocyte size are increased [25]. PVAT, similar to every other adipose tissue, secretes cytokines, hormones, growth factors, and adipokines. It plays a beneficial role as long as adipokine levels with opposing properties remain in equilibrium. In obesity, PVAT becomes dysfunctional and exerts detrimental effects on vascular homeostasis [26].

## 2. The Influence of PVAT-Derived Factors on Vascular Function

Under physiological conditions, PVAT plays a significant role in the regulation of vascular tone, intravascular thermoregulation, and vascular smooth muscle cell (VSMC) proliferation (Figure 1) [27,28,29]. PVAT exhibits an anticontractile effect as a response to several factors such as endothelin-1, phenylephrine, angiotensin II, and serotonin [30,31,32]. PVAT anticontractile factors are divided into adipocytes-derived relaxing factors (ADRF) and perivascular-derived relaxing factors (PDRF) [30,31,32,33]. On the other hand, PVAT induces vasoconstriction by releasing angiotensin II [34] and the superoxide anion [35]. These factors affect vascular tone via endocrine and paracrine mechanisms. Moreover, VSMCs play a significant role in maintaining the balance between vasoconstriction and vasodilator signals. However, PVAT, as a special adipose tissue, is not only a mechanical support for the vasculature but plays a vital role in the homeostasis of the vascular system, sharing a status no less important than that of the endothelium [36].

### 2.1. Adiponectin

Adiponectin is profusely produced and released by PVAT under physiological conditions [37]. It is secreted in different polymeric forms, which differ from each other by molecular weight: multimeric, hexameric, trimeric, and globular [38]. Adiponectin is known as a vasodilator, which acts through different mechanisms affecting endothelial cells and VSMCs directly. First of all, adiponectin activates in both mentioned locations 5′ adenosine monophosphate-activated protein kinase (AMPK), which is responsible for phosphorylation of endothelial nitric oxide synthase (eNOS) [39,40]. Moreover, adiponectin increases the production of 3,4-tetrahydrobiopterin (BH4), which is an important cofactor of eNOS [41]. As a consequence, the synthesis of the well-known vasodilator nitric oxide (NO) is increased. In addition, AMPK induces the opening of large-conductance calcium-activated potassium channels (BK channels) in VSMCs, which also contributes to vasodilation [42]. AMPK is activated mainly by globular and trimeric forms of adiponectin. Additional effects of multimeric adiponectin are the downregulation of glucose blood levels by the stimulation of glucose uptake by muscles, the improvement of insulin sensitivity, or a reduction in hepatic glucose production [43]. Moreover, it also regulates fatty acid metabolism by increasing the high-density lipoprotein (HDL) concentration and decreasing the triglycerides [44].

### 2.2. Leptin

Leptin is another adipokine highly produced by adipose tissue. It regulates appetite by centrally affecting the hypothalamus and activating a sympathetic effect by affecting the arcuate nucleus in the hypothalamus [45]. Leptin synthesis is directly proportional to adipocyte size [46]. Leptin-induced vasodilation acts via endothelium-dependent and -independent mechanisms, which additionally depends on the type of vessel. In large arteries, such as the aorta, leptin increases endothelium-dependent vasodilation, in a similar way to adiponectin, by AMPK activation, which is responsible for eNOS phosphorylation [47]. In small arteries, such as the mesenteric artery, leptin-induced increased synthesis of NO and endothelium-derived hyperpolarizing factor (EDHF) contribute to endothelium-dependent vasodilation [48]. VSMCs are also known as a target for leptin, which impairs the contraction effect of angiotensin II by reducing the Ca^2+^ release from cellular reserves and inducing VSMC proliferation [47]. In addition, leptin in a high concentration can induce vasoconstriction by increasing the endothelin-1 release from endothelium [49]. Leptin is known as an immune-modulatory factor, which induces the production of proinflammatory cytokines such as IL-6, TNFα, or IL-12 and the differentiation of monocytes into macrophages [50].

### 2.3. H_2_O_2_

PVAT-derived hydrogen peroxide (H_2_O_2_) is known as both a vasoconstrictor and vasodilator factor by different mechanisms, which depend on the concentration of H_2_O_2_, vessel type and vessel contractile status [51]. In healthy individuals, the concentration of H_2_O_2_ is nontoxic. First of all, H_2_O_2_ has membrane-permeable properties and freely diffuses to smooth muscle cells, where it stimulates the soluble guanylyl cyclase (sGC-1), which plays the role of receptor for NO in smooth muscles and induces vasodilation by the NO/sGC-1/cGMP pathway [52]. On the other hand, a high concentration of H_2_O_2_ stimulates cyclooxygenase and increases the level of Ca^2+^ [51].

### 2.4. H_2_S

Hydrogen sulfide (H_2_S) is a gaseous factor, which is produced by PVAT, endothelial cells, and VSMCs, controlling the vascular tone. H_2_S-induced vasodilation is caused by activation of BK channels in VSMCs, which leads towards cell membrane hyperpolarization, inactivation of voltage-dependent l-type Ca^2+^ channels, and a decrease in intracellular Ca^2+^ concentration [53]. In addition, H_2_S leads to a dose-dependent decrease in intracellular pH, which causes the vasodilation. It is suggested that the Cl^−^/HCO_3_^−^ ionic exchanger is engaged in this process [54]. The shortage of H_2_S is important in the development of various cardiovascular diseases, such as hypertension, atherosclerosis, and heart failure [55].

### 2.5. Angiotensin 1–7 and Angiotensin II

Components of the renin–angiotensin–aldosterone system (RAAS) are present in the aortic and mesenteric PVAT, except renin [56]. The effect of factors on vascular tone is different. Angiotensin 1–7 induce vasodilation by endothelium-dependent mechanisms. After the activation of the Mas receptors, located in the endothelium, the synthesis of NO is increased, which leads to vasodilation by the activation of BK channels [57]. On the contrary, angiotensin II, which is also produced by PVAT, induces vasoconstriction. There are regional differences in angiotensin II synthesis by PVAT, while it is greater in mesenteric adipose tissue than in the periaortic adipose tissue [35]. Moreover, angiotensin II in increased concentration activates immune cells, which can produce cytokines and proinflammatory mediators [58].

### 2.6. NO

Nitric oxide (NO) is a well-known endogenous gas with vasodilative properties, which is produced in almost all human cells. There are three isoforms of NOS: neuronal NOS (nNOS), inducible NOS (iNOS), and endothelial NOS (eNOS) [59]. Each of them is characterized by different attributes. nNOS is present in cells of the central and peripheral nervous system, where produced NO acts as a neurotransmitter and plays a role in the central regulation of blood pressure [60]. iNOS is activated by inflammatory cytokines and plays a role in inflammation; additionally, iNOS is Ca^2+^ independent, unlike other isoforms [59]. eNOS, which is located in endothelial cells, regulates blood pressure locally and has an antiatherosclerotic effect [60,61]. PVAT is responsible for increased production of NO by a direct mechanism, while eNOS isoform is also present in PVAT, where NO is directly produced and released affecting vasculature [62]. In addition, NO produced in PVAT positively regulates adiponectin release by PVAT [37]. On the other hand, PVAT-derived factors, mentioned in previous sections, increase NO production, which is responsible for activations of BK channels and stimulating cGMP synthesis endothelium and smooth muscle cells.

### 2.7. COX-Derived Factors

PVAT is also known as a source of a group of adipose-derived factors, which are produced by cyclooxygenase (COX), such as thromboxane A2 (TXA2), prostaglandin D2, prostaglandin E2, prostaglandin, F2a, prostaglandin H2, and prostaglandin I (prostacyclin) [63,64]. Both COX-1 and COX-2 isoforms are present in adipocytes of PVAT [64]. COX-derived factors have a different effect on vascular tone. Under normal conditions, serotonin stimulates the production of TXA2, which is characterized by pro-contractile and pro-inflammatory properties [65]. Prostacyclin is known as a vasodilator, which additionally reacts with receptors on the vasculature and plays a significant role in protection against endothelium dysfunction and atherosclerosis [27,66,67]. Moreover, the concentration of prostacyclin declines with age progression [27].

### 2.8. Noradrenaline

PVAT is known as a source of noradrenaline (NA), which is the main neurotransmitter in the sympathetic nervous system (SNS). NA induces vasoconstriction of vessels by activation of adrenoreceptors localized on VSMCs [68]. NA is released by the endings of sympathetic nerves, which are spread among adipose tissue. However, it is reported that NA is additionally synthesized by PVAT independently of SNS activation, while surgical denervation of PVAT decreased NA concentration insignificantly [69,70]. Moreover, Ayala-Lopez et al. also indicated that NA is stored in PVAT, while tyramine stimulation induced an increased release of catecholamines from PVAT. In addition, NA stimulates the synthesis of H_2_O_2_ by PVAT, which has an anti-contractile effect [71].

## 3. The Role of Inflammation, Oxidative Stress, and Hypoxia in Obesity

Obesity is characterized by an excessive level of triglycerides and lipids, which are stored in adipocytes. It leads to their hyperplasia and hypertrophy, where hyperplasia is a well-tolerated complication. In contrast, it is suggested that the capacity of lipid storage and subsequent growth in adipocyte size is limited, and exceeding this threshold induces serious molecular changes and induces cellular dysfunction and death of adipocytes [72]. Moreover, enlarged adipocytes induce elevation of IL-6, IL-8, and leptin and decrease the level of adiponectin, which leads to consequent accumulation of inflammatory factors in PVAT [73,74]. Cytokines, fatty acids, and cell-free DNA, which are excreted after adipocytes apoptosis, induce migration of macrophages to the adipose tissue. Adipose tissue macrophages are divided into two subgroups, which differ from each other by type of secreted cytokines and cell markers: M1 with an inflammatory profile and M2 with an immunosuppressive feature [75]. The M1 subclass secretes cytokines such as TNF-α, IL-6, and IL-1β and plays a significant role in inducing an inflammatory state in adipose tissue, which is important in the development of vascular disorders. Obesity is accompanied by a chronic low-grade inflammatory state, which is confirmed by an elevated level of inflammatory markers, especially C-reactive protein and IL-6, which are significantly higher among obese nonmorbid patients and positively correlates with BMI [76,77].

The excess of carbohydrates, fatty acids, and hyper nutrition induce oxidative stress activation by various pathways such as glycoxidation, oxidative phosphorylation in mitochondria, and NADPH oxidase (NOX) activation with consequent reactive oxygen species (ROS) production [78,79]. The increase in NOX activity leads to excessive production of the superoxide anion (O_2_^−^), which can react with DNA, lipids, and proteins leading to their destruction [80]. Moreover, O_2_^−^ leads to the alteration of NO activity and consequent endothelial dysfunction and cardiovascular events among obese populations [81]. In addition, an elevated level of ROS induces VSMC proliferation and remodeling, which contribute to hypertension development and increased risk of cardiovascular events [82,83].

Moreover, hypertrophy of adipocytes does not proceed hand in hand with angiogenesis, and the demand of tissues for oxygen is greater than the supply. As a result, hypoxia and consequent necrosis and inflammation occur [84]. Hypoxia-inducible factor (HIF-1α), which is increased in adipose tissue among obese individuals, plays the role of mediator in hypoxia. HIF-1α induces the elevation of IL-6 and TNF-α activity and reduces adiponectin concentration [85].

## 4. Dysregulation of Vascular Function Induced by PVAT Dysfunction in Obesity

PVAT dysfunction, which is a result of an increased inflammatory state among obese patients, induces the dysregulation of vascular function by increased peripheral resistance and vascular tone [86]. In animal models, with diet-induced obesity, loss of the PVAT anticontractile effect was correlated with increased blood pressure [87,88]. The exact mechanism of vascular dysfunction in obesity is still not well clarified; however, there are some pathways such as renin–angiotensin–aldosterone system (RAAS) disorders and PVAT-derived factors’ dysregulation, which are involved in the process. Pathological modifications in the synthesis and secretion of ADR and PDRF are responsible for the loss of their anticontractile effect (Figure 2). Endothelial dysfunction plays a crucial role in the pathophysiological process of microvascular and macrovascular complications of obesity [89,90].

### 4.1. The RAAS in Obesity

The renin–angiotensin–aldosterone system (RAAS) plays a significant role in the regulation of blood pressure. Components of RAAS are produced by the adrenal gland, liver, and adipose tissue. The hypertrophy of adipocytes among obese patients contributes to increased production of angiotensinogen, Ang-II, and aldosterone by PVAT and subcutaneous adipose tissue [91,92]. The concentration of factors mentioned above positively correlates with BMI among obese nonhypertensive subjects [93]. Moreover, the animal model study with diet-induced obese rats revealed the increased production of angiotensinogen by adipose tissue, while the liver expression remained unchanged [94], which shows the significant role of adipose tissue in RAAS factor production. Each component of RAAS plays an important role in the development of dysfunction of microcirculation, hypertension, and arterial stiffness. Increased concentrations of Ang-II and aldosterone induce microvascular constriction via different mechanisms. First of all, chronic activation of angiotensin-II type 1 receptors (AT1R), which suppresses eNOS activity, induces a decrease in NO concentration and bioavailability [95]. Moreover, Ang-II stimulates the synthesis and release of endothelium-dependent vasoconstrictors such as endothelin-1 by increased expression of preproendothelin-1, which is consequently converted to endothelin-1 and COX-1-derived prostanoids by increased expression of COX in human endothelial cells [96,97]. In addition, each component of RAAS, especially aldosterone, by the activation of mineralocorticoid receptors, induces reabsorption of sodium in distal nephron and consequently increases blood pressure [98]. Ang-II and aldosterone also induce arterial stiffening via enhanced fibrosis, proliferation of VSMCs, and increased collagen deposits [99].

### 4.2. PVAT-Derived Factors Dysregulation with Effect on Vascular Tone

Among obese patients, the regulatory effect of PVAT on vascular tone is attenuated because of the dysregulation in PVAT-derived factors release. Increased inflammation in adipose tissue, especially IL-6 and TNFα, decreases adiponectin secretion, while the expression of mRNA remains unchanged. Almabrouk et al. reported that concentration of adiponectin was 70% decreased among mice, which were fed a high-fat diet for 12 weeks [100]. Aghamohammadzadeh et al. presented similar findings among humans, while the adiponectin level was significantly lower among obese patients compared to healthy individuals; however, the main limitation of this study was the small study group [101]. Peroxisome proliferator-activated receptor gamma (PPAR-γ) belongs to ligand-dependent nuclear receptors, which play a significant role in adipocyte differentiation and metabolism [102]. Obesity induces downregulation of PPAR-γ, which decreases the expression of adiponectin [103]. Decreased synthesis and secretion of adiponectin is responsible for the decrease in anticontractile effect.

Leptin synthesis is proportional to adipose size; thus, in obesity, the production of leptin is increased in PVAT and visceral adipose tissue [104]. An increased level of leptin is related to selective insensitivity in appetite and weight regulation and a biphasic effect on vascular function. Even though leptin has an anticontractile effect by inducing NO synthesis, long-lasting exposure of endothelium to leptin induces the inverse result by decreasing the bioavailability of NO [105]. Initially, in diet-induced obesity in animal models, the impairment in leptin-induced NO synthesis and release was compensated by enhanced EDHF-mediated vasodilation. After three months of a high-fat diet, both NO and EDHF-mediated vasodilation effects were reduced and led to an increase in blood pressure [44,48]. In addition, an increased level of leptin decreased the level of L-arginine, resulting in eNOS uncoupling and overproduction of O_2_^−^, which reacted with NO forming the cytotoxic ONOO-inducing endothelial dysfunction [105]. Increased leptin levels predispose the development of atherosclerosis, by inducing the production of IL-6, IL-12, and TNFα, which plays a significant role in atherogenesis [106]. Moreover, leptin induces the proliferation of vascular and endothelial cells, which also play an important role in atherosclerosis development [107]. In addition, increased blood pressure induced by an enhanced level of leptin is a result of the SNS stimulation. In experimental studies, the external infusion of leptin increased the concentration of norepinephrine and epinephrine dose dependently [107]. Harlan et al. suggested that leptin induces SNS stimulation by activation of the arcuate nucleus in hypothalamus [108].

Geng et al. showed that H_2_S bioavailability and synthesis in adipose tissue was decreased in mice on a highfat diet [109]. It is probably a result of the decreased activation of cystathionine γ-lyase (CSE), which is responsible for H_2_S production. Moreover, obesity is related to an increased concentration of ROS, which induces H_2_S degradation [110]. A decreased concentration of H_2_S results in increased systolic and diastolic blood pressure and contributes to the progression of atherosclerosis [111]. H_2_O_2_ is another vasorelaxant factor, which has the opposite abilities to O_2_^−^, which induces vasoconstriction. The final effect on vessels depends on the balance between them and the activation of superoxide dismutase (SOD), which is an essential antioxidant enzyme in inactivation of the mentioned factors [25]. The oxidative stress, present among obese individuals, reduces SOD activation, which results in the loss of anticontractile effect and consequent vascular dysfunction by increased concentration of O_2_^−^ and H_2_O_2_ [40]. It turns out that H_2_O_2_ is changed into a hydroxyl radical, which induces cell damage.

Decreased NO release is present among obese patients compared to controls [56]. First of all, disturbances in secretion of the PVAT-derived factors mentioned above induces changes in NO synthesis and releasing. Elevated ROS concentration among obese individuals reacts with the NO forming cytotoxic product, which interferes with endothelial cells. Among diet-induced obese mice, reduced expression of eNOS in mesenteric PVAT [87] and impaired eNOS function in thoracic aorta PVAT [49] was observed.

Obesity is connected with increased COX-1 and consequent COX-2 activation, affecting increased production of contractive (TXA2), which contributes to an increased vascular smooth muscles tone [64]. Moreover, due to pro-inflammatory TXA2, the increased concentration of inflammatory markers, ROS, and consequent endothelial dysfunction is present [112,113]. HF diet attenuates prostacyclin secretion, which induces endothelial dysfunction [27]. Moreover, obese subjects present impairment in the prostacyclin ability to increase adenosine 3′,5′-cyclic monophosphate (cAMP) and consequent dysregulation in platelets function [114].

## 5. The Influence of Exercise among Obese on PVAT

Exercise is one of the beneficial nonpharmacological interventions, which is essential in obesity treatment. Physical activity increases the muscles’ demands for oxygen and nutrition, resulting in increased blood flow and the vasodilation of muscle vessels. Exercise induces body weight loss, improvement in endothelial function, and reduction in blood pressure [115]. The influence of physical activity on PVAT is multiple. First of all, exercise has anti-inflammatory properties and reduces oxidative stress [116]. It was also shown that exercise prevents or attenuates infiltration of immune cells into PVAT improving vascular function [117]. It contributes to decreasing adipocyte size and consequent reduction in inflammatory markers and pro-inflammatory factors secretion such as TNFα, IL-6, or leptin [118,119]. Physical activity contributes to enhanced adiponectin synthesis by PVAT, inducing an improvement in endothelium-dependent vasodilation and vascular function [120]. Moreover, exercise training is responsible for a reduction in ROS concentration and increased NO bioavailability [121]. In obese rats, physical activity increases the eNOS expression and phosphorylation, which is essential in enzyme activation [120,122]. SNS modulates vascular tone via a double mechanism. Noradrenaline, which is the main SNS mediator, induces the expression of β3-adrenoreceptors, which are responsible for adiponectin release, while organic cation transporter 3 (OCT3) separates the excess of NA [38]. Saxton et al. reported that obesity contributes to the downregulation of mentioned receptors among mice [123]. However, exercise performed on obese mice turned out to increase the expression of β3-adrenoreceptors and returned control levels of OCT3. Moreover, stimulation of β3-adrenoreceptors plays an important role in eNOS activation in PVAT [124]. Additionally, it was postulated that skeletal muscle activity regulates PVAT function through myokines such as FGF21, meteorin-like, irisin, IL-15, and IL-6, acting in a paracrine fashion to antagonize dysfunction of PVAT (i.e., inflammation and dysregulated secretion of adipokines) [117]. DeVallance et al. showed that 8-week aerobic training in rats with metabolic syndrome prevented the increase in oxidant load and inflammation, while enhancing •NO and proteasome function in PVAT, which favorably influenced the function of aortic endothelium [125]. Wang et al. reported that among obese mice, the dominant population of macrophages in the PVAT are M1 subtype, which has proinflammatory properties [126]. Physical activity has an influence on the population of macrophages in PVAT, while 8-week exercise induces reduction in M1 cells with rising of M2 cells among exercised mice, which was not observed among the obese nonexercised group of mice [126]. Moreover, mentioned study confirms the thesis presented by the remaining reports that physical activity has anti-inflammatory properties, while 8-week exercise induces a significant reduction in IL-6 and TNF-α concentration with a consequent significant increase in adiponectin and IL-10 level. Uncoupling protein 1 (UCP1), especially highly expressed in BAT, is responsible for heat production by distracting the proton gradient in mitochondria [127]. Exercise intervention induces upregulation of expression of UCP1 and increases the number of multilocular brown adipocytes in PVAT among obese, exercised mice [126]. Similar findings were reported by Liao et al., who reported an increase in UCP1 expression in the mesenteric artery PVAT after exercise intervention among rats [128].

## 6. The Potential Influence of Diet on PVAT-Derived Factors among Obese Patients

Besides exercise, an adequate diet is another beneficial intervention in weight loss and obesity treatment. Nowadays, there are a wide variety of diet strategies, which differ from one another in terms of the percentage content of macronutrients, such as carbohydrates, proteins, and fats. However, a reduction in daily calorie intake is a universal rule and a recommended strategy in weight loss [129]. Some data that present the influence of dietary intervention directly on PVAT are available. Nevertheless, the association between diets and PVAT are not clearly understood. Reports mainly refer to animal models, in which high-carbohydrate (HC) diets induce obesity and the consequent loss of the anticontractile effect of PVAT by an imbalance in PVAT-derived factor secretion [100]. In contrast, Costa et al. have shown that consuming an HC diet for 4 weeks enhanced the release of vasodilatory factors from PVAT, suggesting that this could be a compensatory adaptive characteristic in order to preserve the vascular function during the initial stages of obesity [130]. Additionally, it is suggested that imbalanced diets can cause PVAT inflammation and dysfunction as well as impaired vascular function. The recent published study has showed that a high-fat (HF) and a high-sucrose (HS) diet affected PVAT at different sites. Sasoh et al. have presented characteristic differences in the effects of HF and HS diets on PVAT and aortae [131]. A HF diet induced an increased number of large-sized lipid droplets and increased cluster of differentiation (CD) 68+ macrophage- and monocyte chemotactic protein (MCP)-1-positive areas in the abdominal aortic PVAT (aPVAT). Furthermore, a HF diet caused a decreased collagen fiber-positive area and increased CD68+ macrophage- and MCP-1-positive areas in the abdominal aorta. In contrast, a HS diet induced an increased number of large-sized lipid droplets, increased CD68+ macrophage- and MCP-1-positive areas, and decreased UCP-1 positive area in the thoracic aortic PVAT (tPVAT). Moreover, a HS diet caused a decreased collagen fiber-positive area and increased CD68+ macrophage- and MCP-1-positive areas in the thoracic aorta. However, there were some factors that did not follow the trend to this variation. For example, angiotensinogen levels were increased in both tPVAT and aPVAT of the HF group. The authors concluded that the potential mechanisms underlying these effects may be related to the different adipocyte species that comprise tPVAT and aPVAT [131]. Victorio et al. reported that the effect of HF and HS diets on PVAT differs depending on sex [132]. The anti-contractile effect of PVAT was measured by comparing the phenylephrine-induced contraction in mesenteric arteries after 3 and 5 months of HF or HF+HS diet among male and female mice. The results showed that anticontractile function was impaired after 3 months of both obesogenic diets among females, while among males, the anti-contractile effect remained comparable during the experiment. Moreover, the assessment of PVAT-derived endothelial function after acetylcholine administration likewise demonstrated differences between sexes, while obesogenic diet among females induces endothelial dysfunction after 3 months and only after 5 months among males.

However, there are many reports that relate the positive impact of different diets on inflammatory state, oxidative stress, NO, adiponectin, or leptin concentration. Thus, one could conclude that similar changes could be observed in PVAT; however, further studies should be conducted.

The Mediterranean diet (MD) is the most popular diet and is commonly known as a healthy, balanced diet with proven efficiency in reducing the cardiovascular risk among high-risk patients and reducing overall mortality [133,134]. A typical MD contains 55–60% carbohydrates, mainly complex ones, 25–30% polyunsaturated and monounsaturated fats, and 15–20% proteins, and meals are generally based on fish, nuts, olive oil, and plant-based foods [135]. Luisi et al. reported that the implementation of an MD for 3 months among overweight/obese patients, with high-quality extra virgin olive oil, induced weight loss and the significant elevation of adiponectin levels [136]. Interestingly, among normal weight controls, the MD has no impact on weight, and the increase in adiponectin concentration was not as considerable as that found among overweight/obese patients. It can be concluded that weight loss and the consequent reduction in adipose tissue contribute to a size reduction in adipocytes and an improvement in adiponectin synthesis and release. Among both groups, the concentration of IL-6 significantly decreased after dietary intervention, which provides proof of the anti-inflammatory properties of MD [136]. Moreover, it is suggested that the higher the amount of fiber in one’s diet, the greater the adiponectin concentration in one’s blood [137].

Recently, the ketogenic diet has become very popular due to its therapeutic properties in relation to different diseases. It has been widely used in drug-resistant epilepsy with good outcomes and is increasingly being used in metabolic disorders such as obesity or diabetes mellitus [138,139]. The ketogenic diet is characterized by low carbohydrates and high fat, inducing changes in the metabolism of energy substrates, with a switch from glucose to fatty acids [138]. A very low-calorie ketogenic diet (VLCKD) is a special type of caloric reduction diet characterized by a very low or extremely low daily food energy intake, circa 800 kcal per day [140]. It provides 30–50 g of carbohydrates, about 30–40 g of fats, and 0.8–1.5 g/kg of ideal body weight (IBW) of proteins [141]. Monda et al. reported that obese patients who consumed a VLCKD diet for 8 weeks presented with a significant body mass reduction, a decreased concentration of inflammatory markers such as IL-6, TNF-α, and CRP, and a significant elevation in the level of adiponectin in their blood [140]. This relatively short period of intervention induced a significant multifactorial improvement; however, the main limitation of the aforementioned study is the small sample size. In other reports, the ketogenic diet has also been proven to have anti-inflammatory properties [142,143].

A low-calorie diet is a balanced diet with a 20–30% reduction in one’s daily calorie intake, differing from the ketogenic diet by its macronutrient content, and consisting of 45–55% carbohydrates, 25–35% fat, and 15–25% proteins, with an additional 30 g of fiber [144]. The beneficial effects of a calorie reduction among humans include weight loss, a reduction in superoxide and inflammatory factor production, and the upregulation of the activity of eNOS [145,146]. Vink et al. found that 12 weeks of a low-calorie diet among obese patients resulted in a significant reduction in leptin and the elevation of the adiponectin concentration [147]. Another report among overweight patients confirmed a reduction in body weight and leptin levels after 6 months of a low-calorie diet [148]. It is reported that the supplementation of melatonin in animal models, which is known for its anti-inflammatory and anti-oxidative properties, could be used in the prevention of obesity [149,150]. Szewczyk-Golec et al. reported that 30 days of daily oral administration of 10 g of melatonin, accompanied by a low-calorie diet, induced a statistically significant reduction in body mass weight and led to the elevation of the adiponectin concentration among obese patients, which was not observed among the control group, who received a low-calorie diet alone [151].

Intermittent fasting (IF) is the type of diet that is based on taking intermittent breaks from eating. There are two types of IF: alternative day fasting and time-restricted fasting. The first type includes 24 h of fasting followed by a 24-h period of eating with mixing fast days with nonrestricted days during a week, while time-restricted fasting consists of different variations, f.e., 16 h of fasting with 8 h of eating [152]. IF is reported to have a positive impact on adipose tissue function especially among obese individuals. Liu et al. reported that eight weeks of IF among high-fat diet (HFD)-fed mice reduced adipocytes hypertrophy and concentration of inflammatory markers by bodyweight reduction and improvement in insulin sensitivity [153]. Moreover, IF is more effective in reducing inflammatory markers concentration than a calorie-restriction diet among animal models [154]. Interestingly, the comparison between IF and a calorie-restricted diet among humans showed an intermittent increase in M1 markers of inflammation, which was a result of lipolysis and consequent increase in non-esterified fatty acids serum concentration after IF [155]. On the other hand, IF in mice was reported to induce activation of macrophages with M2 subtype polarization [156]. The M2 macrophages, which are known in the literature for their anti-inflammatory properties, induce the production of IL-10, phagocyte apoptotic cells, and promote wound healing [157,158]. Moreover, fasting-mediated activation of AMPK is reported among animal models [159] and seems to play a significant role in mitochondrial homeostasis [160]. In addition, AMPK, opposite to mTOR, turned out to activate autophagy, which seems to play a significant role in maintaining autophagic homeostasis [161]. Permanent overnutrition induces suppression of autophagy and the accumulation of impaired cellular components, such as mitochondria, in metabolic tissues, which contributes to metabolic dysfunction and consequent development of metabolic disorders such as obesity [162].

## 7. Conclusions

PVAT, as a special adipose tissue, is not only a mechanical support for the vascular system but plays a vital role in the homeostasis of the vascular system, sharing a status no less important than that of endothelium. Obesity plays an important role in the development of vascular dysfunction, dysregulation of vascular tone, and endothelial dysfunction. The pathogenesis of obesity contains hypoxia, increased oxidative stress, and enhanced inflammatory factors. All these components induce PVAT dysfunction, dysregulation in the synthesis of PVAT-derived factors, decreased bioavailability of NO, an increased inflammatory state in PVAT, and increased activation of RAAS. Exercise training, commonly known as an essential nonpharmacological intervention in obesity treatment, contributes to an improvement in PVAT activity, which could have a positive effect on vascular tone. Diet, which is the second most beneficial non-invasive strategy in obesity treatment, may have a positive impact on PVAT-derived factors and may restore the balance in their concentration. Independent of the type of diet, a decrease in body mass weight, which is connected with a reduction in adipose tissue, may restore the balance of the synthesis and release of adipokines. However, further studies should be conducted in order to demonstrate the exact influence of diet on PVAT among humans. Moreover, additional studies are also needed to help researchers better understand the pathophysiology of PVAT and evaluate whether targeting PVAT function could be used as a novel approach for the treatment of cardiovascular diseases.

## Figures and Tables

**Figure 1 nutrients-13-03843-f001:**
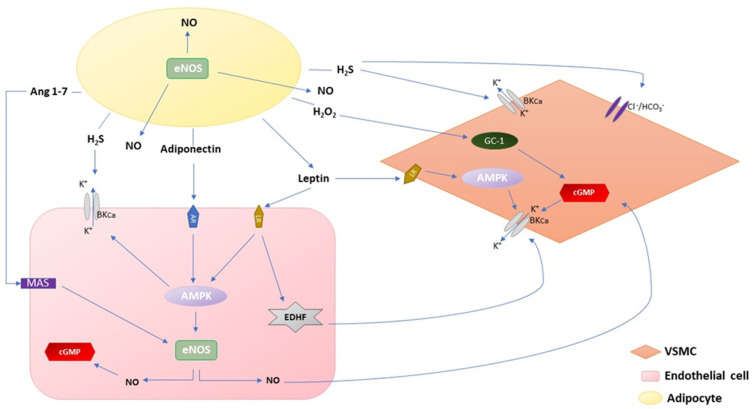
Vasodilatory and hyperpolarizing effect of perivascular adipose tissue–derived factors (PVAT-derived factors) on vascular smooth muscle cells (VSMCs) and hyperpolarizing effect of PVAT-derived factors on endothelial cells. Adiponectin causes vasodilation by affecting adiponectin receptors (AR) in endothelial cells, which contributes to the activation of locations 5′ adenosine monophosphate-activated protein kinase (AMPK), which is responsible for the activation of endothelial NO synthase (eNOS). Enhanced NO concentration induces activation of cyclic guanosine monophosphate (cGMP), which is responsible for opening large-conductance calcium-activated potassium channels (BKCa). eNOS is present in both endothelial cells and adipocytes. Leptin activates leptin receptors (LR), which are responsible for activation of not only AMPK, but also endothelium-derived hyperpolarizing factor (EDHF), which activates BKCa. Moreover, AMPK independently activates BKCa and induces a hyperpolarization effect. Hydrogen sulfide (H2S) induces activation of BKCa in VSMCs and endothelial cells. Moreover, it induces a decrease in intracellular pH by the activation of Cl^−^/HCO3^−^ ionic exchanger. Angiotensin 1–7 (Ang 1–7) by affecting endothelial Ang 1–7 receptor (MAS) activates eNOS and increases the NO concentration. Hydrogen peroxide (H_2_O_2_) stimulates the soluble guanylyl cyclase (sGC-1), which induces vasodilation through the NO/GC-1/cGMP pathway.

**Figure 2 nutrients-13-03843-f002:**
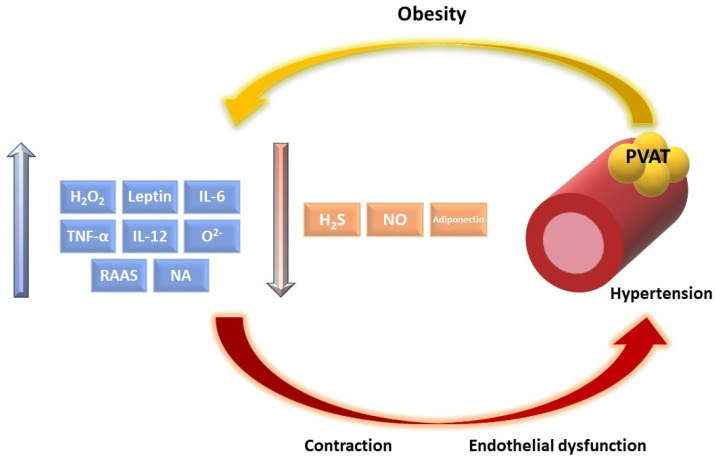
The influence of obesity on PVAT-derived factors’ synthesis resulting in vascular dysfunction. (PVAT: perivascular adipose tissue, RAAS: renin–angiotensin–aldosterone system, NO: nitric oxide, H_2_S: hydrogen sulfide, H_2_O_2_: hydrogen peroxide, TNF–α: tumor necrosis factor α, and NA: noradrenaline).

## Data Availability

We used PubMed and web of science to screen articles for this narrative review. We did not report any data.

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
