# Peer review of "The Role of Obesity-Induced Perivascular Adipose Tissue (PVAT) Dysfunction in Vascular Homeostasis"

_nutrients, 2021, doi:10.3390/nu13113843_

Round 1

Reviewer 1 Report

Much of the manuscript has already been reviewed many times, and in much more depth, in particular see PMID: 31339053. Both figures in this manuscript are similar to figures in PMID: 31339053.

Sections 5 and 6 are promising. If the authors focus on expanding on these sections, that would make the manuscript stand out as different to the already published literature on PVAT. There are a number of recent and relevant publications that could be included, such as PMID: 33687595, PMID: 33425218, PMID: 32893373, PMID: 34566644, PMID: 33929898, PMID: 33897419.

Author Response

Dear Reviewer,

Thank you very much for the review of our work and your valuable comments. Undoubtedly, taking them into account will improve quality of this article. All your remarks have been included in the manuscript.

Changes in the body text are written in red font.

Much of the manuscript has already been reviewed many times, and in much more depth, in particular see PMID: 31339053. Both figures in this manuscript are similar to figures in PMID: 31339053.

Authors`reply: Thank you for your comment. In our opinion, the article raises important issue, while we tried to assemble information about physiology of PVAT, pathophysiology in obesity and the consequent impact on vascular function and the influence of non-invasive lifestyle modifications on PVAT function. From our point of view, Figure 1 presents biochemical pathways of PVAT-derived factors and their influence on vessels and endothelium, which are universal and equal, therefore figure from our article may resemble figures and graphs from other articles, which raise the same issue, however they are not the same. However, we decided to modify Figure 2 to avoid similar assumption.

Sections 5 and 6 are promising. If the authors focus on expanding on these sections, that would make the manuscript stand out as different to the already published literature on PVAT. There are a number of recent and relevant publications that could be included, such as PMID: 33687595, PMID: 33425218, PMID: 32893373, PMID: 34566644, PMID: 33929898, PMID: 33897419.

Authors` reply: In accordance with Reviewer`s remarks we expanded section 5 and 6. We included the all mentioned publications.

Please read the attachment. 

Reviewer 2 Report

The authors report an interesting review on the role of obesity-induced PVAT dysfunction in vascular homeostasis. First, the authors explore the modulatory role of PVAT-derived factors on vascular function, specifically on vascular tone regulation. The authors then explore how obesity impairs PVAT dysfunction, focusing on the role of inflammation, oxidative stress and hypoxia, followed by a section on vascular impairment induced by PVAT dysfunction in obesity. The authors close out the topic by exploring strategies to prevent PVAT dysfunction associated with obesity, namely exercise and diet.

Overall, the manuscript appears to provide a well-balanced review on the topic of interest. However, the following issues must be addressed:

  1. Review articles have been recently published on PVAT, obesity and vascular homeostasis (e.g. Man et al. 2020, Antioxidants, 9:574, doi: 10.3390/antiox9070574; Queiroz and Sena 2020, Ageing Res Rev, 59:101040, doi: 10.1016/j.arr.2020.101040). The authors should provide an explanation on what distinguishes this manuscript from the others.
  2. When talking about the modulatory role of PVAT-derived factors on vascular function (section 2), the authors focus their attention on adiponectin, leptin, hydrogen peroxide, hydrogen sulfide, angiotensin and NO. However, there are other PVAT-derived factors which have not been mentioned, namely thromboxane A2 (Mendizabal et al., 2013, doi 10.1016/j.lfs.2013.10.021; Meyer et al., 2013, doi 10.1371/journal.pone.0079245) and prostacyclin (Chang et al., 2012, doi 10.1161/CIRCULATIONAHA.112.104489; Mendizabal et al., 2013, doi 10.1016/j.lfs.2013.10.021), which play an important role in the modulation of vascular tone. Therefore, the authors should seek to include them in this review, as well as other PVAT-derived factors, e.g. noradrenaline.

Additional issues:

  1. In line 93, the authors state “The degree of AMPK activation depends on the type of adiponectin isoforms”. This concept should be further explored.
  2. Terminology should be consistent with the majority of the literature. Therefore, the sGC abbreviation should be used for designating soluble guanylyl cyclase instead of GC-1. Similarly, superoxide anion is commonly referred as O2.- and not as SO2. Also, peroxynitrite is commonly known as ONOO- (line 277).

Minor points:

L. 89: tetrahydrobiopterin should be replaced by 3,4-tetrahydrobiopterin (BH4)

L. 140: Ca2+ should be replaced by Ca2+

L. 190: TNF-a should be replaced by TNF⍺

L. 191: IL-1B should be replaced by IL-1β

L. 346: The sentence that begins with "Sasoh et al." lacks the number of the reference

L. 358: The sentence “The Authors have been concluded (..)” should be revised

Author Response

Dear Reviewer,

Thank you very much for the review of our work and your valuable comments. Undoubtedly, taking them into account will improve quality of this article. All your remarks have been included in the manuscript.

Changes in the body text are written in red font.

The authors report an interesting review on the role of obesity-induced PVAT dysfunction in vascular homeostasis. First, the authors explore the modulatory role of PVAT-derived factors on vascular function, specifically on vascular tone regulation. The authors then explore how obesity impairs PVAT dysfunction, focusing on the role of inflammation, oxidative stress and hypoxia, followed by a section on vascular impairment induced by PVAT dysfunction in obesity. The authors close out the topic by exploring strategies to prevent PVAT dysfunction associated with obesity, namely exercise and diet.

Authors` reply: Thank you for your opinion of our article.

Review articles have been recently published on PVAT, obesity and vascular homeostasis (e.g. Man et al. 2020, Antioxidants, 9:574, doi: 10.3390/antiox9070574; Queiroz and Sena 2020, Ageing Res Rev, 59:101040, doi: 10.1016/j.arr.2020.101040). The authors should provide an explanation on what distinguishes this manuscript from the others.

Authors` reply: Thank you for your remark. Undoubtedly, mentioned articles are undeniably good source of knowledge about PVAT. However, in our opinion PVAT dysfunction is very important issue, because of all metabolic and cardiovascular disorders which are connected with PVAT disabilities. In our article, we focused on exercise and diet, which are non-invasive cheap, widely available methods. We decided to include information about the influence of different types of diet, which are commonly used in the treatment of obesity or bodyweight reduction. We showed that mentioned diets are connected with positive impact on PVAT function, especially when they result in weight loss. We would like to underline the meaning of lifestyle modifications, which restore the PVAT function with positive impact on endothelium, vascular function and consequent reduction of cardiovascular risk.

When talking about the modulatory role of PVAT-derived factors on vascular function (section 2), the authors focus their attention on adiponectin, leptin, hydrogen peroxide, hydrogen sulfide, angiotensin and NO. However, there are other PVAT-derived factors which have not been mentioned, namely thromboxane A2 (Mendizabal et al., 2013, doi 10.1016/j.lfs.2013.10.021; Meyer et al., 2013, doi 10.1371/journal.pone.0079245) and prostacyclin (Chang et al., 2012, doi 10.1161/CIRCULATIONAHA.112.104489; Mendizabal et al., 2013, doi 10.1016/j.lfs.2013.10.021), which play an important role in the modulation of vascular tone. Therefore, the authors should seek to include them in this review, as well as other PVAT-derived factors, e.g. noradrenaline.

Authors` reply: Thank you for your remarks. We added information about thromboxane A2, prostacyclin and noradrenaline (section 2) and we hope that it will improve the quality of our article.

In line 93, the authors state “The degree of AMPK activation depends on the type of adiponectin isoforms”. This concept should be further explored.

Authors` reply: We explored our concept in line 92-93.

Terminology should be consistent with the majority of the literature. Therefore, the sGC abbreviation should be used for designating soluble guanylyl cyclase instead of GC-1. Similarly, superoxide anion is commonly referred as O2.- and not as SO2. Also, peroxynitrite is commonly known as ONOO- (line 277).

Authors` reply: All the remarks were included in the manuscript.

Minor points:

  1. 89: tetrahydrobiopterin should be replaced by 3,4-tetrahydrobiopterin (BH4)
  2. 140: Ca2+ should be replaced by Ca2+
  3. 190: TNF-a should be replaced by TNF⍺
  4. 191: IL-1B should be replaced by IL-1β
  5. 346: The sentence that begins with "Sasoh et al." lacks the number of the reference
  6. 358: The sentence “The Authors have been concluded (..)” should be revised

Authors` reply: All the remarks were included in the manuscript.

Please read the atachment. 

Round 2

Reviewer 1 Report

The manuscript has been substantially improved. No further comments.

This manuscript is a resubmission of an earlier submission. The following is a list of the peer review reports and author responses from that submission.